# Well-Defined Construction of Functional Macromolecular Architectures Based on Polymerization of Amino Acid Urethanes

**DOI:** 10.3390/biomedicines8090317

**Published:** 2020-08-29

**Authors:** Takeshi Endo, Atsushi Sudo

**Affiliations:** 1Molecular Engineering Institute, Kyushu Institute of Technology, Sensui-cho 1-1, Tobata-ku, Kitakyushu, Fukuoka 804-8550, Japan; 2Department of Applied Chemistry, Faculty of Science and Engineering, Kindai University, Kowakae 3-4-1, Higashi Osaka, Osaka 577-8502, Japan; asudo@apch.kindai.ac.jp

**Keywords:** amino acids, polypeptides, urethane, drug delivery, biosensors

## Abstract

Polypeptide synthesis was accomplished using the urethane derivatives of amino acids as monomers, which can be easily prepared, purified, and stored at ambient temperature without the requirement for special precautions. The urethanes of amino acids are readily synthesized by the *N*-carbamoylation of onium salts of amino acids using diphenyl carbonate (DPC). The prepared urethanes are then efficiently cyclized to produce amino acid *N*-carboxyanhydrides (NCAs). Thereafter, in the presence of primary amines, the ring-opening polymerization (ROP) of NCAs is initiated using the amines, to yield polypeptides with controlled molecular weights. The polypeptides have propagating chains bearing reactive amino groups and initiating chain ends endowed with functional moieties that originate from the amines. Aiming to benefit from these interesting characteristics of the polypeptide synthesis using the urethanes of amino acids, various macromolecular architectures containing polypeptide components have been constructed and applied as biofunctional materials in highly efficient antifouling coatings against proteins and cells, as biosensors for specific molecules, and in targeted drug delivery.

## 1. Introduction

Owing to their potential biological activity, biocompatibility, and biodegradability, polypeptides are attractive building blocks for the construction of a wide variety of macromolecular architectures with various functions. α-Helix and β-sheet, the two typical structural motifs, play important roles in maintaining the three-dimensional (3D) structures of naturally occurring proteins, which are present in various organs, and in enzymes controlling biochemical reactions.

In nature, polypeptides are synthesized in a specific and precise manner according to the corresponding genic information, which defines the sequences of the constituent amino acids. In contrast, synthetic polypeptides are produced by the ring-opening polymerization (ROP) of amino acid *N*-carboxyanhydrides (NCAs) (Scheme 1) [1,2,3]. The ROP of NCAs is initiated using various amines, leading to the synthesis of polypeptides with functionalized terminals, including block copolymers, using polymers having amino terminals as the macroinitiators and graft polymers using polymers bearing amino groups at the side chains and amino-functionalized surfaces. The ROP of NCAs proceeds in a living manner, producing the corresponding polypeptides with predictable molecular weights and narrow molecular weight distributions. Furthermore, the propagating chain end of the formed polypeptides is functionalized with an amino group, which can be used for several chemical conversions, such as the synthesis of macromonomers. Recently, an alcohol-initiated living polymerization of NCAs has also been reported [4].

NCAs can be directly and efficiently synthesized using phosgene (Scheme 2) [5]. In some cases, phosgene derivatives such as trichloromethyl chloroformate and triphosgene can be used as suitable alternatives of phosgene [6,7,8]. However, the NCA synthesis have drawbacks such as the lethal toxicity of the reagents and the evolution of corrosive hydrogen chloride as an unavoidable byproduct. The purification of NCAs is usually difficult because of the labile nature of NCAs, which is also unfavorable for their large-scale production and storage. Thus, the use of NCAs is still limited to experiments in laboratories.

An attractive alternative method for the ROP of NCAs is the use of the sulfur-containing analogs of NCAs (Scheme 3) [9,10,11]. Amino acid *N*-thiocarboxyanhydrides (NTAs), which are synthesized using α-amino acids, undergo ROP in a manner similar to that of NCAs. The use of NTAs for the polypeptide synthesis was reported more than five decades ago [12], and their controlled ROP initiated with primary amines was first reported by Kricheldorf et al. in 2008 [9]. Compared to NCAs, NTAs are more stable under moisture and heat; hence, their isolation and storage are easier than those of NCAs. Moreover, the stability of NTAs under moisture permits their polymerization to be performed in open air [12]. However, there are two limitations in the use of NTAs for the polypeptide synthesis in terms of upscaling. First, dithiocarbonate used for the conversion of amino acids to the corresponding thiourethane derivatives is expensive. Second, the ROP of NTAs is followed by the formation of carbonyl sulfide, which is a toxic gaseous compound.

In contrast, we focus on the development of a more accessible, upscalable, and controllable approach to obtain polypeptides, based on the derivation of amino acids to produce the corresponding urethanes, and their use as key compounds (Scheme 4). In this approach, the urethanes undergo cyclization to produce the corresponding NCAs, and the in situ-formed NCAs undergo ROP to produce polypeptides. In the first step, phenolic compounds are released; however, they do not interfere in the ROP of NCAs. In this short review, we describe the characteristics of the polypeptide synthesis using the urethane derivatives of amino acids and the current scope of its applications in macromolecular architectures.

## 2. Synthesis of Urethane Derivatives of Amino Acids

The urethanes of amino acids bearing the 4-nitrophenoxycarbonyl group in the nitrogen atom can be synthesized directly, by treating the corresponding amino acids with 4-nitrophenyl chloroformate (Scheme 5) [13]. *γ*-Benzyl-L-glutamate, *β*-benzyl-L-aspartate, L-leucine, and L-phenylalanine are efficiently converted to urethanes **1a**–**1d**. Thus, these urethanes **1** are used as the precursors of the corresponding NCAs, which can be isolated or prepared in situ in the polypeptide synthesis.

An alternative synthesis route to *O*-4-(nitrophenyl)urethane derivatives **1** is the reaction of amino acid *tert*-butyl ester with bis(4-nitrophenyl)carbonate (Scheme 6) [10]. The *tert*-butyl group of the resulting urethane is removed using trifluoroacetic acid to produce **1**.

The use of bis(4-nitrophenyl)carbonate in the carbamoylation of amino acids encouraged the use of the diphenyl carbonate (DPC) as a less reactive but considerably less toxic and more cost-effective substitute. Moreover, we focused on the use of DPC because it can be produced using a phosgene-free and environmental-friendly protocol, in which carbon dioxide is used as the source of the carbonyl group of DPC. First, the reaction of amino acids with DPC was attempted; however, no reaction occurred, which is probably due to the suppression of the nucleophilicity of the amino group as a result of amino acid protonation by the carboxyl group [14]. Thus, similar to the synthesis of **1**, the *tert*-butyl esters of amino acids were treated with DPC, to observe if the corresponding urethanes could be synthesized. The finding that the amino group free from protonation is reactive with DPC encouraged the development of a more efficient synthesis route, in which the protection of the carboxyl group by the *tert*-butyl group is not required.

An efficient alternative method to produce phenyl urethanes is shown in Scheme 7. In this method, amino acids are converted to the corresponding tetrabutylammonium salts that possess amino groups released from protonation, hence, it is potentially reactive with DPC [15,16]. The first step is the treatment of amino acids with tetrabutylammonium hydroxide (commercially available as a methanolic solution) in aqueous media. The resulting salts are soluble in a wide range of organic solvents: the nucleophilicity of the amino group and the solubility of the salts allowed the highly efficient reaction of the amino group with DPC to yield the corresponding urethanes **2**. No racemization of the amino acid chiral center is observed. This method enables the efficient conversion of several amino acids bearing alkyl [17], allyl [18], and indolyl [19] groups, as well as protecting hydroxyl [20,21], sulfur-containing [17,22,23], ester [16], amide [20,24], and urethane [25,26,27] groups. It is important to note that L-tryptophan is converted into the corresponding urethane **2f**, which can be used for the synthesis of poly(L-tryptophan) [19]. The conversion of L-tryptophan into an NCA using the conventional method with phosgene is affected by the susceptibility of the indole moiety to hydrogen chloride. In addition, L-serine derived urethane **2u** bearing an acid labile acetal linkage is synthesized and polymerized into the corresponding polypeptides. The acetal linkage is cleaved under acidic conditions to produce poly(L-serine). Urethane **2t** bearing the hydroxyl group [28] and urethane **2u** bearing the acetal moiety [29] are synthesized and polymerized into the corresponding polypeptides.

## 3. Cyclization of Urethane Derivatives to NCAs

Urethanes **1** underwent intramolecular condensation, leading to the formation of NCAs (Scheme 8) [15]. The cyclization of urethane **1a** was monitored by hydrogen-nuclear magnetic resonance (^1^H-NMR), and the nearly completed conversion of **1a** to NCAs within 6 h was confirmed. However, the formed NCAs were unstable under the reaction conditions, resulting in a rapid decrease in the amount of NCAs, owing to their oligomerization. To overcome this limitation, carboxylic acids such as diphenylacetic acid (DPAA) were added to the reaction mixture. Although the addition of DPAA decelerated cyclization, it ensured the stability of NCAs in the reaction mixture, and the NCAs were isolated in high yields.

The cyclization of urethane **2,** which proceeded efficiently at 60 °C, was considerably slower than that of urethane **1**, as expected for the less electrophilic nature of the phenoxy carbonyl group of **2** compared to the 4-nitrophenoxycarbonyl group of **1**. However, the cyclization was significantly promoted by the addition of acetic acid, which prevented the ROP of the resulting NCA (Scheme 9) [16].

## 4. Homopolymerization of Urethane Derivatives into Polypeptides

As described above, the urethane derivatives **1** and **2** undergo cyclization to produce the corresponding NCAs, which can be isolated and used as monomers for polypeptide synthesis. The efficiency of the urethane derivatives as precursors to produce NCAs motivated us to study the ROPs of NCAs prepared in situ using **1** and **2**. The conversion of urethanes into the corresponding polypeptides in a direct manner was achieved by selecting *N*,*N*-dimethylacetamide (DMAc) as the solvent. For example, upon heating the DMAc solution of **1a** at 60 °C, **1a** was consumed completely within 48 h, and the corresponding polypeptide was obtained (Scheme 10) [13]. In this case, the control of the molecular weight was not possible.

Similarly, the urethanes **2** can be converted to the corresponding polypeptide by heating it with DMAc. A typical example is shown in Scheme 11, in which urethane **2s** derived from L-lysine is successfully polymerized [26]. Although this method does not produce polypeptides with controlled molecular weights, it produces polypeptides with high molecular weights.

After ensuring that the urethane derivatives **1** and **2** were the appropriate monomers for the polypeptide synthesis, we focused on the optimization of the conditions that enable the controlled synthesis of polypeptides. The investigation of several additives revealed that the addition of primary amines provided a satisfactory control of the molecular weights of the polypeptides. As observed in Scheme 12, the polymerization using **1a** in the presence of butylamine (1 mol% of the initial amount of **1a**) yielded the corresponding polypeptide, and the molecular weight distribution was considerably less than that obtained by polymerization without the addition of butylamine [30]. The number of average molecular weight *M*_n_ increased proportionally to the feed ratio [**1a**]/[*n*-BuNH_2_]; thus, it was predictable. The analysis of the polypeptides by matrix-assisted laser desorption/ionization (MALDI) time-of-flight (TOF) mass spectrometry (MS) revealed that the selected amine can be introduced into the initiating end of the polypeptide, acting as the ROP initiator of the NCA formed in situ from urethane. The other chain end of the polypeptide is endowed with an amino group, which reacts with the NCA in the propagation step. Moreover, the amino terminal can react with the ester group to form a five-membered lactam moiety at the chain end, leading to the termination of polymerization. The termination is unavoidable at 60 °C, while it can be suppressed by decreasing the polymerization temperature to 30 °C. It is to be noted that polymerization in the absence of butylamine does not proceed at 30 °C, demonstrating the acceleration effect of butylamine on polymerization.

The favorable effects produced by the addition of butylamine motivated us to study the effect of the butylamine addition on the polymerization behavior of urethanes **2**. As shown in Scheme 13, *n*-butylamine was added for the polymerization of **2g** [20]. As a result, the polymerization proceeded efficiently to produce the corresponding polypeptide. The *M*_n_ of the formed polypeptide increases linearly with the increase in feed ratio [**2g**]_0_/[*n*-butylamine]_0_ (Figure 1). Additionally, the molecular weight distribution was maintained at a value less than 1.2. The MALDI-TOF MS analysis of the resulting polypeptides confirmed the incorporation of the amine-derived *N*-butylamino group at the initiating end and the presence of an amino group at the propagating end. These results demonstrate that this polypeptide synthesis is well controlled, and it can be used as a highly efficient alternative approach to the conventional ROP of NCAs.

As observed in the polymerization of **1**, the addition of amine promoted the polymerization and the efficient progress of the polymerization at 30 °C. The acceleration effect was also observed by the addition of amines during the polymerization of **2**. Therefore, we postulated the role of amines as promoters for the cyclization of urethanes **1** and **2** to NCAs. As shown in Scheme 14, the added amine and the amino terminal of the polypeptides allow the proton abstraction of the carboxyl group of **1** and **2**. The resulting carboxylate readily attacks the carbonyl group of the urethane moiety, leading to NCA cyclization and the release of phenols. The ROP of the formed NCAs can be initiated using butylamine, leading to the formation of polypeptide **PP1** bearing a *N*-butylamino group at the initiating end. Meanwhile, the ROP is initiated using NCAs and DMAc, and in these cases, polypeptides bearing an *N*-acylated NCA moiety and an iminium moiety at the initiating ends are produced, respectively. These highly electrophilic initiating ends and the amino group at the propagating end of **PP1** are readily attacked by amines such as butylamine, leading to the formation of **PP1** with increased molecular weights.

Recently, the high tolerance of polymerization to OH-containing compounds has been elucidated (Scheme 15) [28]. Even in the presence of benzyl alcohol, the corresponding polypeptides with predictable *M*_n_s and narrow molecular weight distributions can be obtained. Moreover, the presence of water does not prevent polymerization control. The tolerance of the polypeptide synthesis using the urethane derivatives to alcohols and water demonstrates its potential as a powerful and versatile tool for the construction of a wide range of macromolecular architectures containing polypeptide segments.

## 5. Synthesis of Polypeptides Bearing Functional Side Chains

The high versatility of the polymerization of amino acid urethane derivatives leads to the polymerization of urethanes bearing various functional groups, which produces polypeptides bearing functional side chains that are chemically modifiable. These polypeptides are useful precursors for polypeptide-based functional materials. Polypeptides bearing C–C double bonds in the side chains can be synthesized by the polymerization of **2e** derived from a non-natural amino acid bearing a vinyl group (Scheme 16) [18]. The vinyl group introduced into the side chain reacts with various thiols.

The polymerization of L-cysteine-derived urethane **2m** produces well-defined polypeptides bearing styrene moieties at the side chains (Scheme 17) [23]. The styrene moieties undergo radical polymerization to cross-link the polypeptide chains, producing corresponding 3D-network polypeptides. Additionally, the thiol-ene reaction is performed for cross-linking the polypeptides. In the latter case, the cross-linking is conducted at ambient temperature, allowing for preservation of the polypeptide conformation. Urethane **2m** can be copolymerized with other amino acid-derived urethanes such as L-lysine-derived urethane **2r**, resulting in the density control of the styrenic moieties distributed in the copolypeptides.

Urethane **2q**, which can be synthesized by treating monohydrochloride of L-lysine with DPC in the presence of triethylamine, undergoes polymerization upon the addition of butylamine (Scheme 18) [25]. In the cyclization of **2q** to the corresponding NCA, only the urethane moiety at the *α*-position is used, whereas the urethane moiety at the *ε*-position remains intact. The ROP of the NCA produces a polypeptide in which the side chain inherits the urethane moiety at the *ε*-position. The urethane moiety at the side chain is potentially reactive with amines; hence, it reacts with the amino terminal of the propagating polypeptide, leading to the formation of branched structures and an inadequate control of the molecular weight of the polypeptide. This unfavorable reaction can be prevented by the addition of acetic acid, which converts the amino terminal to its salt, a dormant form that is non-reactive with the urethane moiety. As a result, the target polypeptide is synthesized in a controlled manner. The urethane moiety successfully incorporated into the side chain is used for the reactions with various amines (Scheme 19) [25]. The resulting poly(L-lysine) derivatives are functionalized with the residues originated from the amines that are tethered to the main chain via the urea linkage.

The polypeptide synthesis using urethane derivatives of amino acids can be used in the direct syntheses of polypeptides bearing hydroxyl-functionalized side chains of the corresponding urethanes [28]. For example, L-cysteine-derived urethane **2t** can be converted to the corresponding polypeptide without hydroxyl group protection, because of the tolerance of polymerization to the presence of alcohols (Scheme 20). The versatility of the hydroxyl group as a reactive functional group makes the polymer a useful precursor for functional materials. In contrast, urethane **2u** derived from L-serine does not polymerize, but undergoes cyclization to a chemically robust five-membered urethane (Scheme 21).

## 6. Applications in Macromolecular Architectures

The polymerization of the amino acid urethane derivatives in the presence of amines bearing polymerizable groups produces macromonomers, the polymerization of which produces graft copolymers with polypeptide segments. For example, a macromonomer bearing a styrenic polymerizable chain end and a polypeptide segment can be synthesized by the polymerization of L-methionine-derived urethane **2l** using 4-vinylbenzylamine as a functional initiator (Scheme 22) [22]. The polymerization of **2l** is followed by the acetylation of the terminal amino group, because the present polypeptide-containing macromonomer is designed for the application as an antifouling coating on the surface of hydrophobic polystyrene vessels against biomolecules such as proteins, in which any interaction between the terminal amino group with these biomolecules can be prevented. Then, the resulting macromonomer is completely polymerized into a comb-like polymer, in which the hydrophobic polystyrene-type main chain is adsorbed onto the polystyrene surface in aqueous media for biosensing applications. Thus, the hydrophobic polystyrene surface is successfully covered with highly hydrophilic poly(L-methionine sulfoxide) segments, resulting in low cytotoxicity and excellent antifouling properties against proteins (hRP-IgG) and F9 cells.

π-Conjugated polymers have several characteristics, such as electron conductivity and photoluminescence, and they also perform photoelectric conversion; their combinations with polypeptides lead to the development of various bio-related functional materials. Such combinations of π-conjugated polymers with polypeptides are achieved by the preparation of polypeptides bearing reactive aromatic moieties at the chain ends as macromonomers and the coupling reactions of the aromatic moieties. Polypeptide-type macromonomer **A** (Scheme 23) was prepared by the polymerization of urethane **2s** in the presence of the 1,4-dibromobenzene moiety bearing an amino group [31]. The 1,4-dibromobenzene moiety at the chain end is polymerized into the poly(*p*-phenylene) main chain by the palladium-catalyzed cross-coupling reaction with diboronic acids. Using macromonomer **B** functionalized with the poly(ethylene glycol) chain as a comonomer, the resulting poly(*p*-phenylene) achieved the hydrophilicity that is necessary for the bioelectronic applications of the graft copolymer **C**. Poly(*p*-phenylene) is emissive upon the irradiation of UV light, and its applications in biosensors are investigated.

A similar synthetic approach has been used for the preparation of a polymeric pro-drug for targeted drug delivery (Scheme 24) [32]. The graft copolymer **E** bearing the polythiophene main chain and poly(L-lysine) graft chains was synthesized by the palladium-catalyzed coupling polymerization of the macromonomer **D** bearing the 1,4-dibromothiophene moiety at the terminal, followed by the acid-catalyzed removal of the *t*-butoxycarbonyl group in the side chain. The resulting amino group is used for the attachment of the antitumor drug paclitaxel. Thus, the polypeptide segments are bound by an antibody that acts as a tumor-specific ligand, enabling targeted drug delivery. Furthermore, the π-conjugated and fluorescent poly(thiophene) segments exhibit a radiosensitizer effect, which supports the application of the graft copolymer in bioimaging and radiotherapy. The graft copolymer is not cytotoxic to cells.

Scheme 25 shows the production of a macromolecular architecture that acts as an ethanol biosensor [33]. The polypeptide segment was prepared by the polymerization of L-lysine-derived urethane **2r** using the amino-functionalized bis(3,4-ethylenedioxythiophene) **F**. The resulting macromonomer **G** and bis(thiophene) **H** bearing the ferrocene moieties were electrochemically copolymerized on a graphite electrode, leading to electrode-surface modification by the graft copolymer composed of a π-conjugated main chain, and thus, a conductive main chain, polypeptide graft chains, and redox-responsive ferrocene moieties. The amino group at the chain end of the polypeptide segment was treated with glutaraldehyde to form a tether for the immobilization of the enzyme alcohol oxidase. The produced biosensor is used for the quantitative analyses of ethanol contents in alcoholic beverages.

Scheme 25 shows an example of the electrochemical polymerization of thiophenes endowed with polypeptide chains, which is a suitable and reliable method for depositing the corresponding π-conjugated polymers on the surface of the electrodes. Moreover, the amino terminal of the polypeptide chain is used for the immobilization of various biofunctional molecules, such as glucose oxidase [34], arginylglycylaspartic acid peptide for specific cell recognition [35], and cocaine aptamer for sensing cocaine [36]. These bio-functional molecules bind with specific target molecules, triggering their conformational changes that are transmitted to the π-conjugated polymers through the polypeptide segments. The resultant conformational changes of the π-conjugated polymers are detected as the changes of their electron conductivity.

Scheme 26 shows the cocaine sensors and their compositions [37,38]. On a glassy carbon electrode, π-conjugated polymers bearing poly(L-phenylalanine) graft chains are deposited by the electrochemical polymerization of the corresponding bis(3,4-ethylenedioxythiophene) derivative bearing the polypeptide segment. The benzoylecgonine antibody, which is bound by a cocaine molecule, is attached to the amino terminal of the polypeptide segment [37]. An aptamer can also be used as a site for the recognition of cocaine [38].

Polypeptides containing terminals functionalized with a catechol moiety can be used for surface modification owing to the high affinity of the catechol moiety to a wide range of inorganic materials. Such polypeptides can be easily synthesized by the polymerization of amino acid urethanes using acetonide-protected 3,4-dihydroxybenzylamine, followed by the deprotection of the acetonide group under acidic conditions to produce a catechol moiety at the chain end (Scheme 27) [39,40]. The catechol moiety can be used for the surface modification of glassy carbon electrodes, and the amino group at the chain end of the polypeptide can be used for the attachments of biorecognition elements, such as glucose oxidase [39], anti-immunoglobulin G [39], and antibody [40]. The produced biosensors can be used for enzymatic sensing and affinity sensing.

## 7. Summary

In this paper, we presented a suitable and reliable polypeptide synthesis that uses amino acid derivatives as the monomers. The urethane derivatives function as the precursors for NCAs, which undergo ROP to produce the corresponding polypeptides. In the presence of primary amines, the in situ-formed NCAs undergo ROP in a controlled manner to enable the synthesis of polypeptides, with predictable molecular weights, narrow molecular weight distributions, and well-defined terminal structures, preventing the use of conventional methods for the preparation of amino-acid NCAs. These characteristics of the polypeptide synthesis using amino acid urethanes lead to the precise syntheses of terminal functionalized polypeptides, block copolymers, and graft copolymers containing polypeptide segments.

Currently, the development of biofunctional materials with highly sensitive and selective sensing capabilities for specific bio-related compounds is one of the most important topics of research. The diagnostic and therapeutic applications of biofunctional materials will help improve general health, cure diseases, and prevent infections, resulting in a better quality of life. Synthetic polypeptides will function as indispensable components with biocompatibility and specific 3D shapes for the precise construction of well-defined macromolecular architectures designed for use as biofunctional materials. The polypeptide synthesis strategy using amino acid urethanes will promote the development of macromolecular architectures containing polypeptide segments.

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
