# Peer review of "Well-Defined Construction of Functional Macromolecular Architectures Based on Polymerization of Amino Acid Urethanes"

_biomedicines, 2020, doi:10.3390/biomedicines8090317_

Round 1

Reviewer 1 Report

The review article entitled “Well-defined Construction of Functional Macromolecular Architectures Based on Polymerization of Amino Acid Urethanes” discusses the synthesis of polypeptides starting from amino acid urethane derivatives as monomers. Ring-opening polymerisation of N-carboxyanhydrides (NCAs) will result in the formation of polypeptides with controlled molecular weights. These polypeptide components can be then used in the construction of various macromolecular architectures. The main issue with the current article is that the authors mainly relied on discussing only their own contribution to the field while ignoring the work done by all other research groups working on the same topic (i.e. Out of the 37 references included, 26 references (70%) belong to the authors). Further, many claims within the article have been made by the authors without any supporting references. Some English editing might be also required. All in all, I would not recommend publication in Biomedicines at this stage.

Author Response

Thank you for reviewing our manuscript. In the field of polypeptide synthesis, the method described in the manuscript is relatively new, and there are only a few reports by other research groups. We cited all of their reports. No intention to neglect other groups' works.

According to the suggestion, we cited some references, although they are not directly relevant. The revised manuscript got English proofreading. We hope the revised manuscript meet the standard of the journal.  

Reviewer 2 Report

The authors reported review article presenting 

Author Response

Thank you for reviewing our manuscript. We added some references related to other research groups and the revised manuscript got English proofreading. 

Round 2

Reviewer 1 Report

Accepted in the current format